# Formation Features of Polymer–Metal–Carbon Ternary Electromagnetic Nanocomposites Based on Polyphenoxazine

**DOI:** 10.3390/polym15132894

**Published:** 2023-06-29

**Authors:** Sveta Ozkan, Valeriy Petrov, Andrey Vasilev, Petr Chernavskii, Mikhail Efimov, Dmitriy Muratov, Galina Pankina, Galina Karpacheva

**Affiliations:** 1A.V. Topchiev Institute of Petrochemical Synthesis, Russian Academy of Sciences, 29 Leninsky Prospect, Moscow 119991, Russia; petrov@ips.ac.ru (V.P.); vasilev@ips.ac.ru (A.V.); chern5@inbox.ru (P.C.); efimov@ips.ac.ru (M.E.); muratov@ips.ac.ru (D.M.); pankina5151@inbox.ru (G.P.); gpk@ips.ac.ru (G.K.); 2Department of Chemistry Lomonosov, Moscow State University, 1-3 Leninskie Gory, Moscow 119991, Russia

**Keywords:** ternary nanocomposites, electromagnetic nanomaterials, polyphenoxazine, carbon nanotubes, Co-Fe particles, IR heating

## Abstract

Novel ternary hybrid polyphenoxazine (PPOA)-derived nanocomposites involving Co-Fe particles and single-walled (SWCNTs) or multi-walled (MWCNTs) carbon nanotubes were prepared and investigated. An efficient one-pot method employing infrared (IR) heating enabled the formation of Co-Fe/CNT/PPOA nanocomposites. During this, the dehydrogenation of phenoxazine (POA) units led to the simultaneous reduction of metals by released hydrogen, yielding bimetallic Co-Fe particles with a size range from the nanoscale (5–30 nm) to the microscale (400–1400 nm). The synthesized Co-Fe/CNT/PPOA nanomaterials exhibited impressive thermal stability, demonstrating a half-weight loss at 640 °C and 563 °C in air for Co-Fe/SWCNT/PPOA and Co-Fe/MWCNT/PPOA, respectively. Although a slightly broader range of saturation magnetization values was obtained using MWCNTs, it was found that the type of carbon nanotube, whether an SWCNT (22.14–41.82 emu/g) or an MWCNT (20.93–44.33 emu/g), did not considerably affect the magnetic characteristics of the resulting nanomaterial. By contrast, saturation magnetization escalated with an increasing concentration of both cobalt and iron. These nanocomposites demonstrated a weak dependence of electrical conductivity on frequency. It is shown that the conductivity value for hybrid nanocomposites is higher compared to single-polymer materials and becomes higher with increasing CNT content.

## 1. Introduction

Ternary hybrid nanocomposites, with their constitutive elements including conjugated polymers, carbon nanomaterials, and magnetic nanoparticles, stand at the forefront of materials for modern technologies [1,2,3,4]. These materials hold potential for various applications, such as microwave absorption [5,6,7,8,9,10], energy storage [11,12,13,14,15,16,17,18], water resource remediation [19,20,21,22], drug delivery [23], corrosion resistance improvement [24], catalysis [25,26,27,28], sensing [29,30,31,32], and tracking electromagnetic environmental pollution [33]. Every component of these ternary nanocomposites significantly influences their cooperative properties. Metal nanoparticles often play a major role, giving nanocomposites distinctive properties. In particular, the nature and composition of these nanoparticles largely dictate the catalytic [34,35], magnetic [36,37], photocatalytic [38], electrochemical [39,40,41], sensory [42], and antibacterial [43] characteristics of nanocomposite materials.

Incorporating carbon nanomaterials, such as carbon nanotubes (CNTs), into polymer matrices and composites has become a common technique in materials science due to the many improvements that these carbon components provide [44,45]. The integration of CNTs contributes significantly to a material’s mechanical properties [46]. CNTs are known for their superior tensile strength and elasticity, attributes that, when embedded within a polymer matrix, increase the composite’s overall durability and resistance to wear and damage. In addition, the superior electrical conductivity of CNTs has attracted considerable interest [47,48]. The introduction of CNTs can significantly enhance the electrical performance of polymer composites, making them ideal for a wide array of applications in the electrical and electronics sectors.

Moreover, the interaction between these components can produce synergistic effects, potentially improving the performance of nanocomposites [49,50]. For instance, the integration of metal nanoparticles and carbon nanotubes can boost electron transfer and offer pathways for charge transportation. This feature can be crucial for applications such as energy storage, where efficient charge flow stands as one of the key requirements.

Regarding the role of conjugated polymers, it is worth noting that they contribute additional mechanical properties, along with imparting thermoplastic and conductive characteristics. Furthermore, the existence of a polymer matrix offers a supportive structure capable of efficiently encapsulating and stabilizing the aforementioned nanoparticles, enhancing the composite’s overall stability.

However, the tendency of CNTs to aggregate presents challenges in achieving their uniform distribution within nanocomposites, despite the presence of a stabilizing polymer matrix [51]. Achieving high dispersion can be accomplished via in situ oxidative polymerization of aromatic amine monomers in the presence of both CNTs and magnetic nanoparticles. Conventionally, to produce these ternary materials, oxidative polymerization is conducted on the surface of pre-prepared magnetic nanoparticles like Fe_3_O_4_, α-Fe_2_O_3_, γ-Fe_2_O_3_, and Co_3_O_4_ [29,33,52,53]. Developing a comprehensive synthetic strategy is key to broadening the range of magnetic nanoparticles in nanocomposites.

A trending approach in the development of ternary nanocomposite synthesis involves the careful selection of appropriate polymers. For instance, employing polyphenoxazine (PPOA), a semi-ladder heterocyclic polymer, could be an efficient strategy for establishing a polymer matrix, thereby promoting uniform distribution of the metal and carbon components. In this study, a one-pot procedure for thermostable ternary nanocomposite synthesis based on PPOA, Co-Fe particles, and CNTs is proposed. Initially, a suspension of single-walled (SWCNT) or multi-walled (MWCNT) carbon nanotubes in a co-solution of PPOA and metal salts in dimethylformamide (DMF) was obtained. The resulting composite precursor was then subjected to heat treatment under infrared (IR) radiation to reduce the metal components and form bimetallic Co-Fe nanoparticles. The phase composition, morphology, and thermal, electrical, and magnetic properties of the ternary nanomaterials were thoroughly examined, with a focus on how these characteristics varied based on the conditions under which they were synthesized.

## 2. Materials and Methods

### 2.1. Materials

Phenoxazine (POA) (97%) and DMF (99%) (both of reagent grade from Acros Organics, Geel, Belgium), toluene (analytical-grade), isopropyl alcohol (extra-pure-grade), Co(CH_3_CO_2_)_2_·4H_2_O (pure-grade), FeCl_3_·6H_2_O (pure-grade), and MWCNT (from MER, USA) were utilized as received. An electric arc discharge technique was used to produce SWCNT from Carbon Chg, Ltd. (Moscow, Russia) with dimensions of d = 1.4–1.6 nm and l = 0.5–1.5 µm. (NH_4_)_2_S_2_O_8_ (analytical grade) was subjected to recrystallization for purification. PPOA and Co-Fe/PPOA were synthesized following the methods developed in [54,55].

### 2.2. Synthesis of Co-Fe/CNT/PPOA

A co-solution of PPOA, Co(OOCCH_3_)_2_·4H_2_O, and FeCl_3_·6H_2_O in DMF containing CNTs (either SWCNTs or MWCNTs) was prepared for the synthesis of Co-Fe/CNT/PPOA. The composition included carbon nanotubes with C_CNT_ = 3 and 10 wt%, cobalt (II) C_Co_ = 5–10 wt%, and iron (III) C_Fe_ = 5–20 wt%, each relative to the weight of the polymer. After preparing the precursor, the solvent was evaporated at ~85 °C. The precursor was then subjected to IR heating in an Ar atmosphere at temperatures ranging between 400 and 600 °C for 2–10 min.

### 2.3. Characterization

A laboratory quartz tube IR furnace equipped with halogen lamps (total power of 24 kW) was employed for the synthesis of Co-Fe/CNT/PPOA nanocomposites [56,57]. The sample-loaded graphite box was positioned in the quartz reactor. 

Metal content was measured using an ICPE-9000 Shimadzu ICP emission spectrometer (Kyoto, Japan) via the inductively coupled plasma atomic emission spectroscopy (ICP-AES) method. The quantitative carbon, hydrogen, and nitrogen contents of the examined samples were established using a Flash 2000 CHNS analyzer (Thermo Fisher Scientific, Waltham, MA, USA). XRD studies were conducted using a Difray-401 X-ray diffractometer (Scientific Instruments Joint Stock Company, Saint Petersburg, Russia) via CrKα radiation (λ = 0.229 nm). A Bruker HYPERION-2000 IR microscope (Bruker, Karlsruhe, Germany) coupled with a Bruker IFS 66v FTIR spectrometer (Karlsruhe, Germany) was utilized to record (ATR) FTIR spectra within the range of 600–4000 cm^−1^ (150 scans, ZnSe crystal). Raman spectra were obtained using a Senterra II Raman spectrometer (Bruker, Karlsruhe, Germany) with a wavelength of 532 nm and power of 0.25 mW. TEM and SEM images were captured using a JEM-2100 transmission electron microscope (JEOL, Akishima, Tokyo, Japan) and a Hitachi TM 3030 scanning electron microscope (Hitachi High-Technologies Corporation, Fukuoka, Japan), respectively. Thermogravimetric analysis (TGA) from 30 to 1000 °C was performed using a Mettler Toledo TGA/DSC1 (Giessen, Germany) in both air and argon atmospheres. A Mettler Toledo DSC823e calorimeter (Giessen, Germany) was utilized for differential scanning calorimetry (DSC) in a nitrogen atmosphere.

The magnetic properties of the obtained nanocomposites were investigated using a vibration magnetometer. The dependence of conductivity on current frequency was studied using an E7-20 precision LCR-meter (MC Meratest, Moscow, Russia) in a frequency range of 25.0 Hz–1.0 MHz. 

## 3. Results

### 3.1. Characterization of Co-Fe/CNT/PPOA Nanomaterials

Ternary hybrid nanocomposites were synthesized using a one-pot method, which involved IR heating of a hybrid precursor composed of PPOA, CNTs, and Co (II) and Fe (III) salts. Since the procedure suggests 600 °C heating, a thermostable polymer is necessary. Consequently, we proposed PPOA as the polymer component due to its proven high thermal stability [55]. Figure 1 presents the chemical structure of PPOA, a semi-ladder heterocyclic polymer, where nitrogen and oxygen atoms participate in the conjugated system. 

During heating, the POA units are dehydrogenated, and hydrogen is released, which contributes to the reduction of metal particles. These processes lead to the formation of hybrid ternary Co-Fe/CNT/PPOA nanomaterials, which is confirmed by the FTIR and XRD data described below.

ATR FTIR is used to show that the IR heating of PPOA in the presence of SWCNTs or MWCNTs and Co(CH_3_CO_2_)_2_·4H_2_O and FeCl_3_·6H_2_O salts leads to the dehydrogenation of POA units with the formation of C=N bonds. In the ATR FTIR spectra of Co-Fe/CNT/PPOA, there is a shift and broadening of the bands at 1587 and 1483 cm^−1^, which correspond to stretching vibrations of ν_C–C_ bonds in aromatic rings (Figure 2).

As the synthesis temperature increases, there is a decrease in the intensity of the absorption bands at 3380 and 3060 cm^−1^, corresponding to stretching vibrations of ν_N–H_ and ν_C–H_ bonds in POA units. Just as in PPOA, the absorption bands at 869 and 836 cm^−1^ are due to out-of-plane bending vibrations of the δ_C–H_ bonds of the 1,2,4-substituted benzene ring. The absorption band at 737 cm^−1^ refers to out-of-plane bending vibrations of the δ_C–H_ bonds of the 1,2-substituted benzene ring of the end groups [55]. The intensity of this absorption band decreases, which indicates that the number of end groups of the polymer chain drops with the temperature increase.

The data obtained from the elemental analysis confirm the dehydrogenation of POA units. As the heating temperature rises, there is a notable decrease in hydrogen content in the Co-Fe/CNT/PPOA nanocomposites. The initial C/H ratio for PPOA is 16.1, which, at 600 °C, increases to 35.6 and 39.5 for Co-Fe/SWCNT/PPOA and Co-Fe/MWCNT/PPOA, respectively, as shown in Table 1. The released hydrogen facilitates the reduction of metals, resulting in the formation of bimetallic Co-Fe particles. The reflection peaks of Co-Fe particles at diffraction scattering angles of 2θ = 69.04° and 106.5° (Cr*K*_α_ radiation) correspond to a solid solution [37,54]. 

To illustrate the changes occurring in the materials during the synthesis process, especially the conversion of the metal phase, XRD analysis was performed on the samples. Figure 3 shows the XRD patterns of the Co-Fe/CNT/PPOA nanocomposites (with Co and Fe contents of 5 and 10 wt%, respectively) obtained at different synthesis temperatures. At 400 °C, reflection peaks corresponding to Co-Fe solid solution particles are registered only for the Co-Fe/SWCNT/PPOA nanocomposite (Figure 3a). In the case of nanocomposites containing MWCNTs, the diffraction pattern reveals intense reflection peaks of Fe_3_O_4_ and β-Co particles (Figure 3b). Peaks of β-Co particles, denoting a cubic face-centered lattice, can be observed at 2θ = 67.7°, 80.3°, while peaks for Fe_3_O_4_ particles are present at 2θ = 45.9°, 54.1°, 63.1°, 74.49°, 90.7°, and 100.8°. Therefore, it is evident that the inclusion of SWCNTs enables the alloying of iron and cobalt at lower temperatures in comparison to when MWCNTs are used in the nanocomposite. 

At temperatures above 400 °C in both series, reflection peaks of bimetallic Co-Fe particles at 2θ = 69.0° and 106.5° are observed, indicating almost complete alloying of both metals. At 500 °C, only the reflection peaks of bimetallic Co-Fe particles are observed in the XRD pattern of Co-Fe/MWCNT/PPOA. However, the Co-Fe/SWCNT/PPOA nanocomposite continues to display traces of Fe_3_O_4_ and β-Co (Figure 3a). As the synthesis temperature rises to 600 °C, the metal phase of both Co-Fe/SWCNT/PPOA and Co-Fe/MWCNT/PPOA nanocomposites is represented only by Co-Fe particles. Therefore, this specific temperature was chosen as the optimal condition for synthesizing the ternary nanocomposites to investigate their magnetic, electrical, and thermal properties.

Further, we investigated the effect of the content of different components on the phase composition of the resulting ternary nanocomposites. Figure 4a shows the XRD patterns of nanocomposites with 10 wt% CNT content. Compared to similar composites containing 3 wt% of CNTs (as shown in Figure 3b), increasing the content of the carbon component has no effect on the phase composition. Only the peak related to the graphite-like structure at 39° (Cr*K*_α_ radiation) becomes more distinct in the case of the composite with MWCNTs. The only observable difference is the more pronounced peak corresponding to the carbon phase at 2θ = 39° for the (002) plane, in the case of the composite with MWCNTs, and its intensity grows with the increase in the MWCNT content. This peak is absent in the XRD patterns of Co-Fe/SWCNT/PPOA, and this is explained by the impossibility of obtaining a diffraction pattern from a single SWCNT plane.

Significantly larger variations are observed when the metal content and ratios are adjusted. The XRD patterns of nanocomposites with a higher proportion of cobalt and iron are represented in Figure 4b. 

With an increase in cobalt content, the XRD patterns exhibit not only reflection peaks from the Co-Fe alloy, but also peaks indicative of the β-Co phase. Given the observed shift in these peaks relative to the values typical of monometallic cobalt, it is reasonable to suggest that the XRD patterns represent the reflection peaks of a β-Co-based alloy phase. Increasing the iron content up to 20 wt% leads to the formation of Fe_3_O_4_ particles due to the excess iron relative to cobalt, along with the formation of a Co-Fe solid solution.

To demonstrate the difference in the carbon structure in the resulting ternary nanocomposites, Raman spectra of the samples are shown in Figure 5. From the Raman spectra, it is evident that CNTs significantly influence the carbon structure of nanocomposites, displaying typical profiles of both single- and multi-walled carbon nanotubes [58,59]. 

Specifically, the nanocomposite spectrum involving SWCNTs exhibits a well-defined G-band, indicative of the ordered in-plane carbon structure. In contrast, the spectrum for the nanocomposite with MWCNTs features a pronounced D-band, which is indicative of the sp^3^-hybridization of carbon atoms. Thus, the I_D_/I_G_ ratio indicating the degree of disorder along the hexagonal plane varies greatly and is 0.06 and 0.81 for composites with SWCNTs and MWCNTs, respectively.

Figure 6 and Figure 7 show TEM and SEM micrographs of the obtained nanocomposites. The interaction of the three components of the composites can be seen.

According to the analysis of the images, a mixture of particles with sizes of 5–30 nm and 400–1400 nm is formed in Co-Fe/CNT/PPOA.

Varying the initial loading of metals and their ratios leads to changes in their actual content in nanocomposites. Accordingly, based on the ICP-AES data, cobalt and iron content are found to fluctuate within the ranges of 4.8–14.5% and 4.2–23.5%, respectively (Table 2).

Figure 8 illustrates the temperature-dependent weight decrease of Co-Fe/CNT/PPOA compared to Co-Fe/PPOA when heated up to 1000 °C, both in the argon flow and in air. The introduction of CNTs into the Co-Fe/PPOA composition has little effect on the thermal characteristics of nanocomposites, although it slightly improves them. 

The weight loss at 122–130 °C is due only to the presence of moisture in the nanocomposites, which is confirmed by the endothermic peak on the DSC curves (Figure 9a). The processes of thermal–oxidative degradation of the Co-Fe/SWCNT/PPOA and Co-Fe/MWCNT/PPOA nanomaterials begin at 370 °C. 

According to the DTG data, the decomposition processes of the Co-Fe/CNT/PPOA nanocomposites in air occur within the range of 340–750 °C, with the maximum at 596 °C (Figure 9b). In air, the Co-Fe/SWCNT/PPOA and Co-Fe/MWCNT/PPOA nanomaterials lose half of their initial weight at 643 and 563 °C, and at 1000 °C, the residues are 25 and 34%, respectively. In an inert medium, a gradual loss of weight is observed in the nanocomposites, and at 1000 °C, the residue is 63–70%. In the DTG curves, the maxima at 736 and 774 °C are associated with the reduction of Fe_3_O_4_ residue (Table 3).

Table 3 summarizes the thermal analysis data and thermal characteristics of the nanocomposites. As can be seen, at C_Co_ = 10 wt% and C_Fe_ = 10 wt% upon loading, in Co-Fe/CNT/PPOA nanocomposites compared to Co-Fe/PPOA, there remain traces of Fe_3_O_4_ due to incomplete reduction to Co-Fe. Apparently, part of the released hydrogen is used for the reduction of oxygen-containing groups on the CNT surface. The Co-Fe/SWCNT/PPOA and Co-Fe/MWCNT/PPOA nanomaterials are characterized by high thermal stability.

The advancement in the preparation of these nanomaterials paves the way for high-performance magnetic nanocomposites with improved thermal and electrical properties.

### 3.2. Electromagnetic Properties of Co-Fe/CNT/PPOA Nanomaterials

The dependence of magnetization on the value of the applied magnetic field is shown in Figure 10. Table 2 shows the values of the main magnetic properties of Co-Fe/CNT/PPOA nanocomposites.

As seen in Table 2, the introduction of CNTs into the Co-Fe/PPOA structure results in an increase in saturation magnetization. Presumably, this is connected to the formation of smaller magnetic particles due to a decrease in their aggregation via their attachment to the CNT surface and dispersion in the polymer matrix. The magnetic characteristics of nanomaterials are practically independent of the nature of carbon nanotubes and are as follows for Co-Fe/SWCNT/PPOA and Co-Fe/MWCNT/PPOA, respectively: residual magnetization *M_R_*: 0.25–0.91 emu/g and 0.20–1.00 emu/g, coercive force *H_C_*: 50–75 Oe and 25–49 Oe, saturation magnetization *M_S_*: 22.14–41.82 emu/g and 20.93–44.33 emu/g. The *M_R_*/*M_S_* ratio is 0.009–0.034 for Co-Fe/SWCNT/PPOA and 0.010–0.023 for Co-Fe/MWCNT/PPOA, which indicates the formation of both superparamagnetic and ferromagnetic particles. This means that the magnetic behavior of the ternary nanocomposites is determined by the presence of small superparamagnetic and large ferromagnetic particles in their structure.

The dependence of electrical conductivity on the frequency of the Co-Fe/SWCNT/PPOA and Co-Fe/MWCNT/PPOA nanocomposites compared to Co-Fe/PPOA was studied (Figure 11).

Table 4 shows the values of conductivity (σ_ac_) of the materials at 25 Hz and 1 MHz. Nanomaterials are characterized by a weak dependence of electrical conductivity on frequency, which indicates that the percolation threshold is exceeded. At low frequencies, the electrical conductivity of the nanocomposites is 3.16 × 10^−2^ S/cm (Co-Fe/SWCNT/PPOA) and 1.41 × 10^−1^ S/cm (Co-Fe/MWCNT/PPOA).

The metallic component (Co-Fe) contributes significantly to the conductivity of nanomaterials, because neat PPOA demonstrates a low conductivity value (9.73 × 10^−10^ S/cm). The electrical conductivity of nanocomposites also depends on the concentration of carbon nanotubes. With an increase in the CNT content from 3 to 10 wt%, the electrical conductivity grows from 3.16 × 10^−2^ S/cm to 5.94 × 10^−1^ S/cm for Co-Fe/SWCNT/PPOA and from 1.46 × 10^−1^ S/cm to 7.28 × 10^−1^ S/cm for Co-Fe/MWCNT/PPOA. Thus, the increase in the electrical conductivity of nanomaterials occurs due to the presence of both Co-Fe particles and CNTs.

## 4. Conclusions

The proposed one-pot method enables the preparation of thermostable electromagnetic ternary hybrid nanomaterials via IR heating. New ternary nanocomposites were synthesized under the conditions of the IR heating of PPOA in the presence of CNTs and Co (II) and Fe (III) salts. Under IR heating, the formation of Co-Fe/CNT/PPOA nanomaterials proceeds with the simultaneous dehydrogenation of phenoxazine units and the reduction of metals by evolved hydrogen with the formation of Co-Fe particles with sizes of 5–30 nm and 400–1400 nm. The one-pot method for the formation of hybrid Co-Fe/CNT/PPOA nanomaterials makes it possible, without subjecting the polymer matrix to degradation, to obtain magnetic bimetallic particles directly during nanocomposite synthesis. These PPOA-derived nanocomposites display magnetic characteristics that exhibit minimal dependence on the type of employed carbon nanotube. Their saturation magnetization was found to be within a range of 22.14–41.82 emu/g for Co-Fe/SWCNT/PPOA and 20.93–44.33 emu/g for Co-Fe/MWCNT/PPOA. It was shown that the conductivity value for hybrid nanocomposites is significantly higher compared to the PPOA single-polymer. Both the metallic component (Co-Fe) and CNTs make a substantial contribution to the nanomaterials’ electrical conductivity. As the CNT content increases from 3 to 10 wt%, a growth in electrical conductivity is noted, escalating from 3.16 × 10^−2^ S/cm to 5.94 × 10^−1^ S/cm for Co-Fe/SWCNT/PPOA, and from 1.46 × 10^−1^ S/cm to 7.28 × 10^−1^ S/cm for Co-Fe/MWCNT/PPOA. In addition, the incorporation of CNTs into the Co-Fe/CNT/PPOA structure has a negligible impact on the thermal stability of the nanocomposites. When exposed to air, the Co-Fe/SWCNT/PPOA and Co-Fe/MWCNT/PPOA nanomaterials demonstrate a half-weight loss at 643 °C and 563 °C, respectively. Under inert conditions at 1000°C, 63–70% of the original weight is retained. These Co-Fe/CNT/PPOA nanomaterials demonstrate potential for deployment in the fabrication of electromagnetic shielding and corrosion-resistant coatings, and the manufacturing of electrode materials for electrochemical devices. 

## Figures and Tables

**Figure 1 polymers-15-02894-f001:**
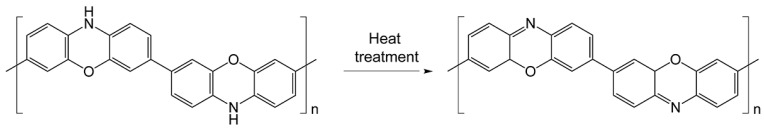
Chemical structure of PPOA and its thermal transformation.

**Figure 2 polymers-15-02894-f002:**
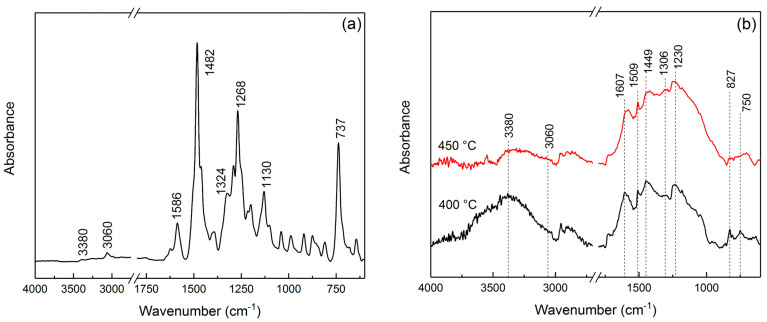
Attenuated total reflection (ATR) FTIR spectra of (**a**) PPOA and (**b**) Co-Fe/CNT/PPOA depending on the synthesis temperature.

**Figure 3 polymers-15-02894-f003:**
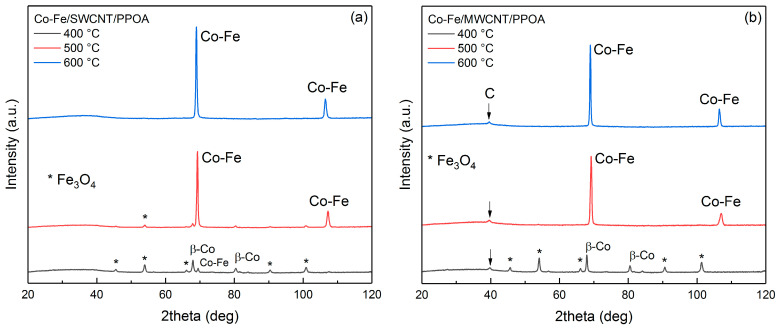
XRD patterns of (**a**) Co-Fe/SWCNT/PPOA and (**b**) Co-Fe/MWCNT/PPOA at 400, 500, and 600 °C. CNT content is 3 wt%. Co and Fe contents are 5 and 10 wt%, respectively.

**Figure 4 polymers-15-02894-f004:**
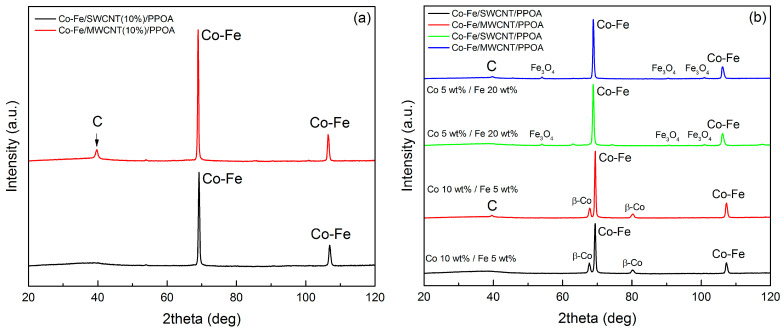
XRD patterns of Co-Fe/CNT/PPOA depending on (**a**) the CNT content (C_CNT_ = 10 wt%, C_Co_ = 5 wt%, C_Fe_ = 10 wt%) and (**b**) metals (C_CNT_ = 3 wt%).

**Figure 5 polymers-15-02894-f005:**
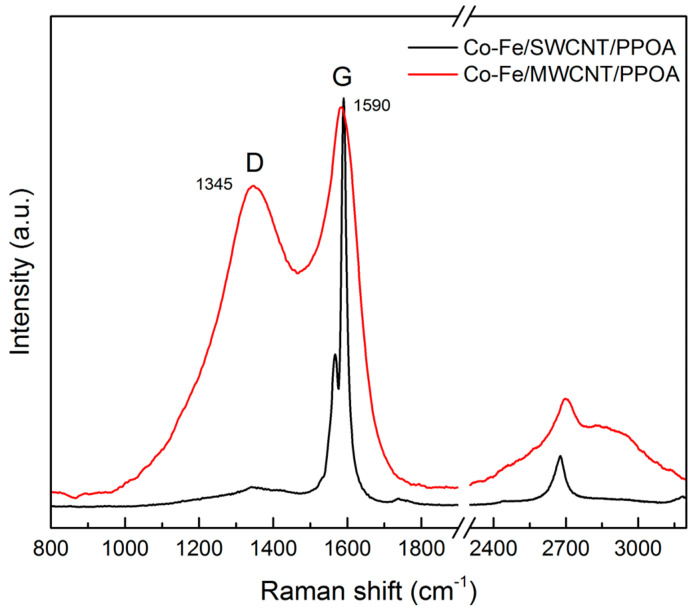
Raman spectra of Co-Fe/SWCNT/PPOA and Co-Fe/MWCNT/PPOA.

**Figure 6 polymers-15-02894-f006:**
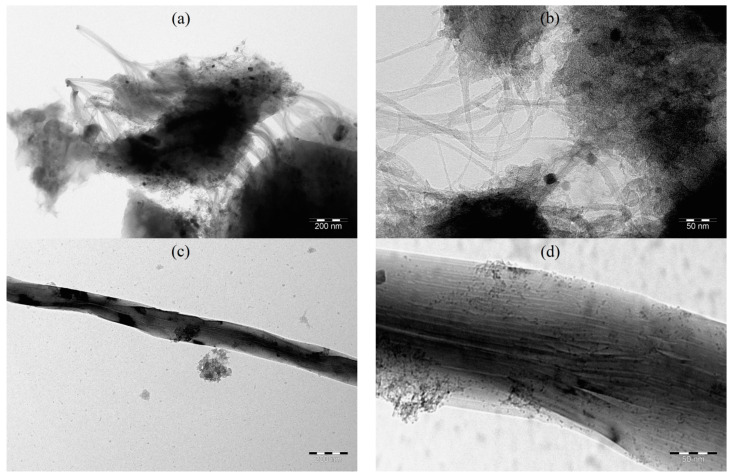
TEM images of (**a**,**b**) Co-Fe/SWCNT/PPOA and (**c**,**d**) Co-Fe/MWCNT/PPOA.

**Figure 7 polymers-15-02894-f007:**
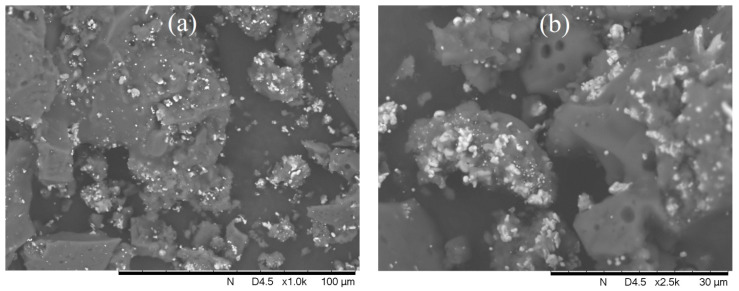
SEM images of (**a**,**b**) Co-Fe/PPOA, (**c**,**d**) Co-Fe/SWCNT/PPOA, and (**e**,**f**) Co-Fe/MWCNT/PPOA.

**Figure 8 polymers-15-02894-f008:**
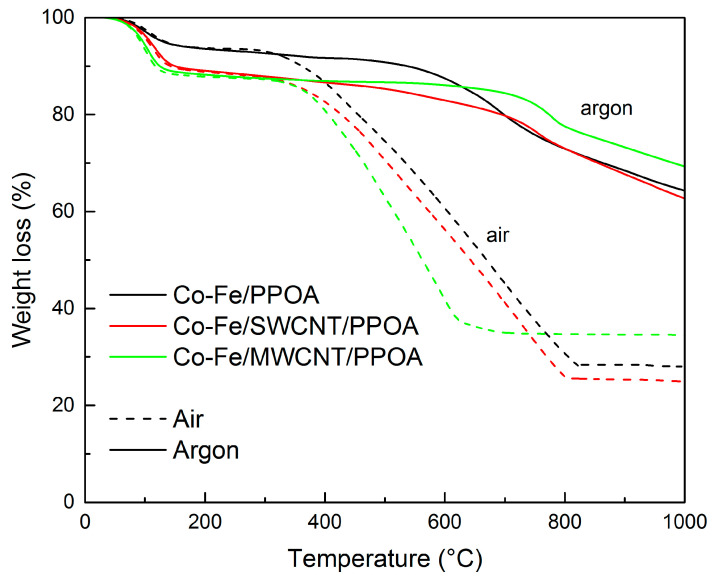
TGA curves of Co-Fe/PPOA, Co-Fe/SWCNT/PPOA, and Co-Fe/MWCNT/PPOA upon heating up to 1000 °C in air and in the argon flow.

**Figure 9 polymers-15-02894-f009:**
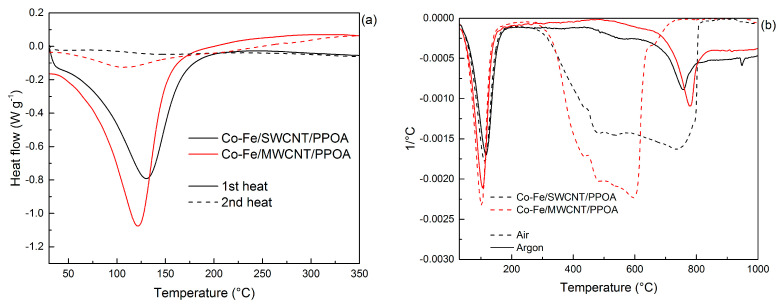
(**a**) DSC (after one and two rounds of heating) and (**b**) DTG curves (in air and in argon flow) of Co-Fe/SWCNT/PPOA and Co-Fe/MWCNT/PPOA.

**Figure 10 polymers-15-02894-f010:**
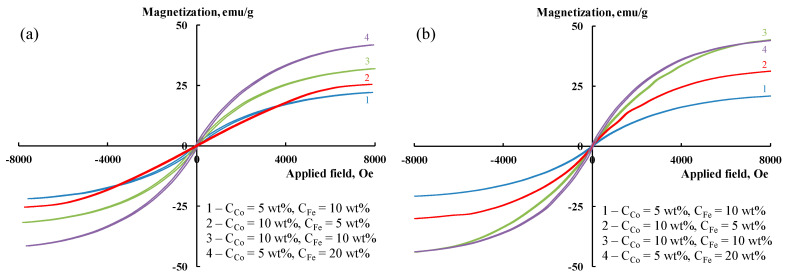
Magnetization of (**a**) Co-Fe/SWCNT/PPOA and (**b**) Co-Fe/MWCNT/PPOA as a function of applied magnetic field.

**Figure 11 polymers-15-02894-f011:**
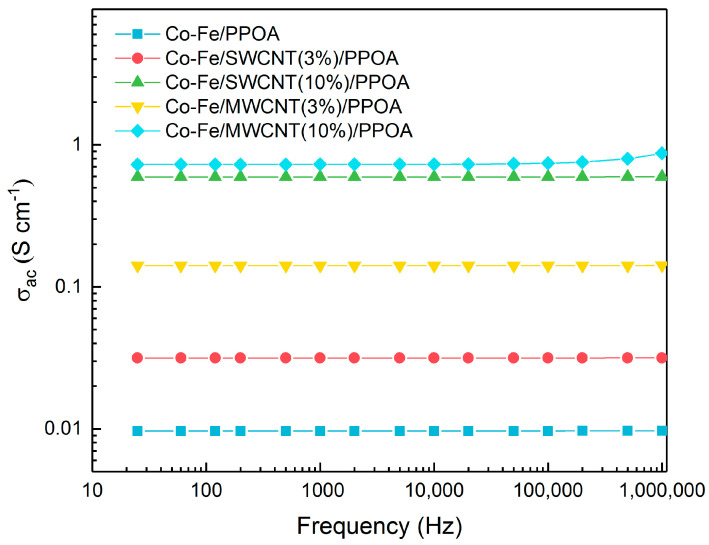
The dependence of conductivity on frequency for Co-Fe/PPOA, Co-Fe/SWCNT/PPOA, and Co-Fe/MWCNT/PPOA nanocomposites.

**Table 1 polymers-15-02894-t001:** ICP-AES data of nanocomposites and elemental analysis data of PPOA component.

Materials	Co, %	Fe, %	C, %	N, %	H, %	O, %	C/N	C/H
PPOA	-	-	78.7	7.7	4.9	8.7	10.2	16.1
Co-Fe/PPOA	11.4	10.3	53.5	5.5	1.8	17.5	9.7	29.7
Co-Fe/SWCNT/PPOA	10.2	7.8	56.9	4.8	1.6	18.7	11.9	35.6
Co-Fe/MWCNT/PPOA	14.5	14.1	55.3	4.3	1.4	10.4	12.9	39.5

C_Co_ = 10 wt% and C_Fe_ = 10 wt% upon loading. C_CNT_ = 10 wt%. Nanocomposites were prepared via IR heating at 600 °C.

**Table 2 polymers-15-02894-t002:** Magnetic properties of nanomaterials.

Nanomaterials	C_Co_ *, wt %	C_Fe_ *, wt %	C_Co_ **, %	C_Fe_ **, %	Co and Fe Phase Composition	*H_C_*, Oe	*M_S_*, emu/g	*M_R_*, emu/g	*M_R_*/*M_S_*
Co-Fe/PPOA	5	10	7.2	12.8	Co-Fe	50	8.96	0.19	0.021
Co-Fe/SWCNT/PPOA	5	10	5.5	9.8	Co-Fe	75	22.14	0.75	0.034
	5	20	4.9	16.3	Co-Fe, Fe_3_O_4_, Fe_4_N	50	41.82	0.91	0.022
	10	5	12.0	5.2	Co-Fe, β-Co	50	25.48	0.25	0.009
	10	10	10.2	7.8	Co-Fe, Fe_3_O_4 (traces)_	64	31.90	0.82	0.026
Co-Fe/MWCNT/PPOA	5	10	6.8	11.1	Co-Fe	32	20.93	0.20	0.010
	5	20	4.8	23.5	Co-Fe, Fe_3_O_4_	37	44.33	0.50	0.011
	10	5	10.9	4.2	Co-Fe, β-Co	25	31.19	0.45	0.014
	10	10	14.5	14.1	Co-Fe, Fe_3_O_4_	49	44.08	1.00	0.023

* Upon loading. ** According to ICP-AES data. C_CNT_ = 10 wt%. Nanocomposites were prepared via IR heating at 600 °C. *H_C_*—coercive force, *M_S_*—saturation magnetization, *M_R_*—residual magnetization.

**Table 3 polymers-15-02894-t003:** Thermal properties of nanomaterials.

Materials	C_Co_ *, %	C_Fe_ *, %	Co and Fe Phase Composition	^∧^ *T*_5%_, °C	^∧∧^ *T*_50%_, °C	^∧∧∧^ Residue, %
PPOA	-	-	-	380/325	580/>1000	0/51
Co-Fe/PPOA	11.4	10.3	Co-Fe	134/131	670/>1000	28/64
Co-Fe/SWCNT/PPOA	10.2	7.8	Co-Fe, Fe_3_O_4 (traces)_	105/109	643/>1000	25/63
Co-Fe/MWCNT/PPOA	14.5	14.1	Co-Fe, Fe_3_O_4_	93/97	563/>1000	34/70

* According to ICP-AES data. C_Co_ = 10 wt% and C_Fe_ = 10 wt% upon loading. C_CNT_ = 3 wt%. Nanocomposites were prepared via IR heating at 600 °C. ^∧^
*T*_5%_, ^∧∧^
*T*_50%_—5 and 50 % weight loss (air/argon), ^∧∧∧^ residue at 1000 °C (air/argon).

**Table 4 polymers-15-02894-t004:** Conductivity values of materials.

Materials	C_CNT_	* σ_ac_, S/cm	
PPOA	0	9.73 × 10^−10^	8.78 × 10^−6^
Co-Fe/PPOA	0	9.67 × 10^−3^	9.71 × 10^−3^
Co-Fe/SWCNT/PPOA	3	3.16 × 10^−2^	3.17 × 10^−2^
	10	5.94 × 10^−1^	5.96 × 10^−1^
Co-Fe/MWCNT/PPOA	3	1.41 × 10^−1^	1.42 × 10^−1^
	10	7.28 × 10^−1^	8.71 × 10^−1^

* Conductivity (σ_ac_) at 25 Hz and 1 MHz.

## Data Availability

Not applicable.

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
