# Peer review of "Formation Features of Polymer–Metal–Carbon Ternary Electromagnetic Nanocomposites Based on Polyphenoxazine"

_polymers, 2023, doi:10.3390/polym15132894_

Round 1

Reviewer 1 Report

polymers-2464812

The authors draw attention to novel ternary hybrid nanocomposites derived from polyphenoxazine (PPOA) incorporating Co-Fe 10 particles and single-walled (SWCNT) or multi-walled (MWCNT) carbon nanotubes were obtained. The synthesized Co-Fe/CNT/PPOA nanomaterials exhibited impressive thermal stability, demonstrating weight loss at 640 °C and 563 °C Although a slightly wider range of saturation magnetization values was obtained. It is shown that the conductivity value for hybrid nanocomposites is higher compared to the monopolymer material and becomes higher with increasing CNT content.

 The proposed topic is innovative and relevant to practice. Manuscript is written in standart English and minor Enslish editing is requered.The authors have described the novelties in detail, using TEM, SEM, Raman spectra analysis, thermal and electromagnetic properties of new matherials. 62% of the references used are from the last 4 years .I have no remarks on the introduction, materials, results in the given form.

Notes:

1)  Black color Selection in indication of the sub-figures in Figure 7.

2) Alignment of the "references" part, according to the requirements (some of the years are in bold; others are not)

3) Conclusion part is small - need to rewrite; some of the main important components of the study missing. 

 Minor editing of English language required

Author Response

The authors are grateful to the reviewer for constructive and valuable comments on the manuscript. Please find below our answers to the comments.

Comments and Suggestions for Authors

1)  Black color Selection in indication of the sub-figures in Figure 7.

Appropriate changes were made.

2) Alignment of the "references" part, according to the requirements (some of the years are in bold; others are not)

Appropriate changes were made.

3) Conclusion part is small - need to rewrite; some of the main important components of the study missing. 

Conclusion part was modified. 

Appropriate additions were introduced into the Conclusion part in colored characters.

Minor editing of English language required

A professional translator has corrected typos and mistakes. Proofreading of the manuscript was carried out.

Reviewer 2 Report

This manuscript study the compositional, electrical and magnetic properties of bimetal particle/CNT/PPOA composite prepared by one-pot method utilizing IR heating. There are several comments for improving the manuscript with some questions to be resolved as below.

1. The captions in Figures are too small. Needs modification.

2. Provide FT-IR spectra in ~ 3000 cm-1 range, so that the statement 'As the synthesis temperature increases, there is a decrease in the intensity of the absorption bands at 3380 and 3020 cm–1, corresponding to stretching vibrations of N–H and C–H bonds in phenoxazine units." could be verified.

3. The most pronounced XRD peak of CNT is at 2 theta ~ 26 degree for (002) plane. Why aren't they observed in the XRD patterns of Figure 3 and 4? SWCNT also shows XRD peak at ~ 26 degree.

4. Usually, the intrinsic electrical conductivity of SWCNT is higher than MWCNT. However, why are the conductivity of Co-Fe/MWCNT/PPOA samples higher compared to Co-Fe/SWCNT/PPOA samples in Figure 11?

N/A

Author Response

The authors are grateful to the reviewer for constructive and valuable comments on the manuscript. Please find below our answers to the comments.

Comments and Suggestions for Authors

  1. The captions in Figures are too small. Needs modification.

Appropriate changes were made.

  1. Provide FT-IR spectra in ~ 3000 cm-1 range, so that the statement 'As the synthesis temperature increases, there is a decrease in the intensity of the absorption bands at 3380 and 3020 cm–1, corresponding to stretching vibrations of N–H and C–H bonds in phenoxazine units." could be verified.

FTIR spectra were provided.

  1. The most pronounced XRD peak of CNT is at 2 theta ~ 26 degree for (002) plane. Why aren't they observed in the XRD patterns of Figure 3 and 4? SWCNT also shows XRD peak at ~ 26 degree.

XRD studies were conducted using a Difray-401 X-ray diffractometer on CrKα radiation (λ = 0.229 nm).

Only the peak related to the graphite-like structure at 39° (CrKα radiation) became more distinct in the case of the composite with MWCNT. The only observable difference was the more pronounced peak corresponding to the carbon phase at 2θ = 39° for (002) plane, in the case of the composite with MWCNT, and its intensity grows with the increase in the MWCNT content. This peak is absent in the XRD patterns of Co-Fe/SWCNT/PPOA, and this is explained by the impossibility of obtaining a diffraction pattern from a single SWCNT plane.

Appropriate additions were introduced into the text in colored characters.

  1. Usually, the intrinsic electrical conductivity of SWCNT is higher than MWCNT. However, why are the conductivity of Co-Fe/MWCNT/PPOA samples higher compared to Co-Fe/SWCNT/PPOA samples in Figure 11?

The higher conductivity of composites including MWCNTs can be determined by a lower percolation threshold. Several interrelated factors, such as the geometric dimensions of nanotubes, the homogeneity of their distribution in the composite, the interaction of distributed CNTs and polymers may influence this parameter. Normally, the degree of percolation for MWCNTs is higher due to the geometric factor, and therefore the electrical conductivity of the composite with MWCNTs is also higher. A more reasoned explanation of the obtained conductivity values requires additional studies that are not part of this work.

Comments on the Quality of English Language

A professional translator has corrected typos and mistakes. Proofreading of the manuscript was carried out.
